# NEURAL DATA FILTER FOR BOOTSTRAPPING STOCHASTIC GRADIENT DESCENT

**Yang Fan** [*]
School of Computer Science and Technology
University of Science and Technology of China
`v-yanfa@microsoft.com`

**Fei Tian & Tao Qin & Tie-Yan Liu**
Microsoft Research
`{fetia,taoqin, tie-yan.liu}@microsoft.com`

## ABSTRACT

Mini-batch based Stochastic Gradient Descent(SGD) has been widely used to train deep neural networks efficiently. In this paper, we design a general framework to automatically and adaptively select training data for SGD. The framework is based on neural networks and we call it *Neural Data Filter* (**NDF**). In Neural Data Filter, the whole training process of the original neural network is monitored and supervised by a deep reinforcement network, which controls whether to filter some data in sequentially arrived mini-batches so as to maximize future accumulative reward (e.g., validation accuracy). The SGD process accompanied with NDF is able to use less data and converge faster while achieving comparable accuracy as the standard SGD trained on the full dataset. Our experiments show that NDF bootstraps SGD training for different neural network models including Multi Layer Perceptron Network and Recurrent Neural Network trained on various types of tasks including image classification and text understanding.

## 1 INTRODUCTION

With large amount of training data as its fuel, deep neural networks (DNN) have achieved state-of-art performances in multiple tasks. Examples include deep convolutional neural network (CNN) for image understanding (Krizhevsky et al., 2012; Ioffe & Szegedy, 2015; He et al., 2015; Ren et al., 2015) and recurrent neural networks (RNN) for natural language processing (Cho et al., 2014; Kiros et al., 2015; Dai & Le, 2015; Shang et al., 2015). To effectively train DNN with large scale of data, typically mini-batch based Stochastic Gradient Descent (SGD) (and its variants such as Adagrad (Duchi et al., 2011), Adadelta (Zeiler, 2012) and Adam (Kingma & Ba, 2014)) is used. The mini-batch based SGD training is a sequential process, in which mini-batches of data $D = \{D_1, \cdots D_t, \ldots, D_T\}$ arrive sequentially in a random order. Here $D_t = (d_1, \cdots, d_M)$ is the mini-batch of data arriving at the $t$-th time step and consisting of $M$ training instances. After receiving $D_t$ at $t$-th step, the loss and gradient w.r.t. current model parameters $\mathcal{W}_t$ are $L_t = \frac{1}{M} l(d_m)$ and $g_t = \frac{\partial L_t}{\partial \mathcal{W}_t}$, based on which the neural network model gets updated:

$$\mathcal{W}_{t+1} = \mathcal{W}_t - \eta_t g_t. \tag{1}$$

Here $l(\cdot)$ is the loss function specified by the neural network and $\eta_t$ is the learning rate at $t$-th step.

With the sequential execution of SGD training, the neural network evolves constantly from a *raw* state to a fairly *mature* state, rendering different views even for the same training data. For example, as imposed by the spirit of Curriculum Learning (CL) (Bengio et al., 2009) and Self-Paced Learning (SPL) (Kumar et al., 2010), at the *baby* stage of the neural network, *easy* examples play important roles whereas *hard* examples are comparatively negligible. In contrast, at the *adult* age, the neural

---

[*]Works done when Yang Fan is an intern at Microsoft Research Asia.

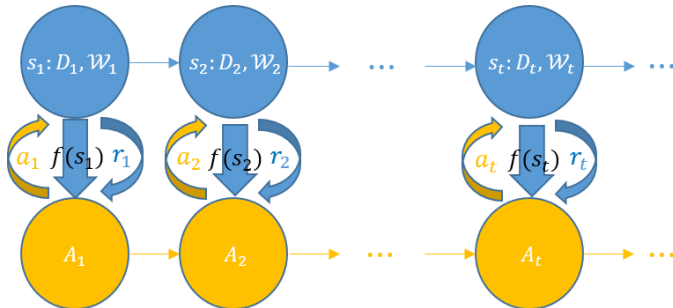

Figure 1: Basic structure of SGD accompanied with NDF. Blue part refers to SGD training process and yellow part is NDF.

network tends to favor *harder* training examples, since *easy* ones bring minor changes. It remains an important question that, *how to optimally and dynamically allocate training data at different stages of SGD training*?

A possible approach is to solve this problem in an *active* manner: at each time step $t$, the mini-batch data $D_t$ is chosen from all the left untrained data (Tsvetkov et al., 2016; Sachan & Xing, 2016). However, this typically requires a feed-forward pass over the whole remaining dataset at each training step, making it computationally expensive. We therefore consider a *passive* way in this paper, in which the random ordering of all the mini-batches is pre-given and maintained during the training process. What actually do is, after receiving the mini-batch $D_t$ of $M$ training instances, we dynamically determine which instances in $D_t$ are used for training and which are filtered, based on the features extracted from the feedforward pass only on $D_t$. Acting in this way avoids unnecessary computational steps on those filtered data and thus speeds-up the training process.

Previous works such as curriculum learning (CL) and self-paced learning (SPL) can be leveraged to fulfill such a data filtration task. However, they are typically based on simple heuristic rules, such as shuffling the sequence length to train language model (Bengio et al., 2009), or abandoning training instances whose loss values are larger than a human-defined threshold (Kumar et al., 2010; Jiang et al., 2014a).

In this work, we propose a Neural Data Filter (NDF) framework from a more principled and self-adaptive view. In this framework, as illustrated in Figure 1, the SGD training for DNN is naturally casted into a Markov Decision Process (MDP) (Sutton & Barto, 1998) and data filtration strategy is fully controlled through deep reinforcement learning (Mnih et al., 2013; Lillicrap et al., 2015b; Mnih et al., 2016). In such an MDP, a state (namely $s_1, \cdots, s_t, \cdots$) is composed of two parts: the mini-batch of data arrived and the parameters of the current neural network model, i.e, $s_t = \{D_t, \mathcal{W}_t\}$. In each time step $t$, NDF receives a representation $f(s_t)$ for current state from SGD, outputs the action $a_t$ specifying which instances in $D_t$ will be filtered according to its policy $A_t$. Afterwards, the remaining data determined by $a_t$ will be used by SGD to update the neural network state and generate a reward $r_t$ (such as validation accuracy), which will be leveraged by NDF as the feedback for updating its own policy.

From another view, while SGD acts as the trainer for base model, i.e., DNN, it meanwhile is the trainee of reinforcement learning module. In other words, reinforcement learning acts at the *teacher* module while SGD for DNN is the *student*. Speaking more ambitiously, such a *teacher-student* framework based on reinforcement learning goes far beyond data filtration for neural network training: On one hand, the base model the can be benefitted is not limited to neural networks; on the other, the action space in reinforcement learning *teacher* module covers any strategies in machine learning process, such as hyper-parameter tuning and distributed scheduling. Through carefully designed interaction between the two modules, the training process of general machine learning models can be more elaborately controlled.

The rest of the paper is organized as follows: in the next section 2, we will introduce the details of Neural Data Filter (NDF), including the MDP language to model Stochastic Gradient Descent training, and the policy gradient algorithms to learn NDF. Then in section 3, the empirical results

of training LSTM RNN will be shown to verify the effectiveness of NDF. We discuss related work in subsequent section 4 and conclude the paper in the last section 5.

## 2 NEURAL DATA FILTER

We introduce the mathematical details of Neural Data Filter (NDF) for SGD training in this section. As a summary, NDF aims to filter certain amount of training data within a mini-batch, in order to achieve better convergence speed for SGD training. To achieve that, as introduced in last section and Figure 1, we cast Stochastic Gradient Descent training for DNN as a Markov Decision Process (MDP), termed as **SGD-MDP**.

**SGD-MDP**: As traditional MDP, SGD-MDP is composed of the tuple $< s, a, \mathcal{P}, r, \gamma >$, illustrated as:

- $s$ is the state, corresponding to the mini-batch data arrived and current neural network state: $s_t = (D_t, \mathcal{W}_t)$.

- $a$ represents the actions space and for data filtration task, we have $a = \{a_m\}_{m=1}^M \in \{0, 1\}^M$, where $M$ is the batch size and $a_m \in \{0, 1\}$ denotes whether to filter the $m^{th}$ data instance in $D_t$ or not[1]. Those filtered instances will have no effects to neural network training.

- $\mathcal{P}_{ss'}^a = P(s'|s, a)$ is the state transition probability, determined by two factors: 1) The uniform distribution of sequentially arrived training batch data; 2) The optimization process specified by Gradient Descent principle (c.f. equation 1). The randomness comes from stochastic factors in training, such as dropout (Srivastava et al., 2014).

- $r = r(s, a)$ is the reward, set to be any signal indicating how well the training goes, such as validation accuracy, or the lost gap for current mini-batch data before/after model update.

- Furthermore future reward $r$ is discounted by a discounting factor $\gamma \in [0, 1]$ into the cumulative reward.

NDF samples the action $a$ by its policy function $A = P_\Theta(a|s)$ with parameters $\Theta$ to be learnt. For example, NDF policy $A$ can be set as logistic regression:

$$A(s, a; \Theta) = P_\Theta(a|s) = a\sigma(\theta f(s) + b) + (1 - a)(1 - \sigma(\theta f(s) + b)), \tag{2}$$

where $\sigma(x) = 1/(1 + \exp(-x))$ is sigmoid function, $\Theta = \{\theta, b\}$. $f(s)$ is the feature vector to effectively represent state $s$, discussed as below.

**State Features**: The aim of designing state feature vector $f(s)$ is to effectively and efficiently represent SGD-MDP state. Since state $s$ includes both arrived training data and current neural network state, we adopt three categories features to compose $f(s)$:

- Data features, contains information for data instance, such as its label category (we use $1$ *of* $|Y|$ representations), the length of sentence, or linguistic features for text segments (Tsvetkov et al., 2016). Data features are commonly used in Curriculum Learning (Bengio et al., 2009; Tsvetkov et al., 2016).

- Neural network features, include the signals reflecting how *well* current neural network is trained. We collect several simple features, such as passed mini-batch number (i.e., iteration), the average historical training loss and current validation accuracy. They are proven to be effective enough to represent current neural network status.

- Features to represent the combination of both data and model. By using these features, we target to represent how *important* the arrived training data is for current neural network. We mainly use three parts of such signals in our classification tasks: 1) the predicted probabilities of each class; 2)the cross-entropy loss, which appears frequently in Self-Paced

---

[1] We consider data instances within the same mini-batch are independent with each other, therefore for statement simplicity, when the context is clear, $a$ will be used to denote the remain/filter decision for single data instance, i.e., $a \in \{0, 1\}$. Similarly, the notation $s$ will sometimes represent the state for only one training instance.

> Learning algorithms  (Kumar et al., 2010; Jiang et al., 2014a; Sachan & Xing, 2016); 3) the
> margin value [2].

The state features $f(s)$ are computed once each mini-batch training data arrives.

The whole process for training neural networks is listed in Algorithm 1. In particular, we take the similar generalization framework proposed in  (Andrychowicz et al., 2016), in which we use part of training data to train the policy of NDF (Step 1 and 2), and apply the data filtration model to the training process on the whole dataset (Step 3). The detailed algorithm to train NDF policy will be introduced in the next subsection.

---

**Algorithm 1** Training Neural Networks with Neural Data Filter.

---

**Input**: Training Data $D$.
1. Sample part of NDF training data $D'$ from $D$.
2. Optimize NDF policy network $A(s; \Theta)$ (c.f. equation 2) based on $D'$ by policy gradient.
3. Apply $A(s; \Theta)$ to full dataset $D$ to train neural network model by SGD.
**Output**: The Neural Network Model.

---

## 2.1    Training Algorithm for NDF Policy

Policy gradient methods are adopted to learn NDF policy $A$. In particular, according to different policy gradient methods, we designed two algorithms: **NDF-REINFORCE** and **NDF-ActorCritic**.

**NDF-REINFORCE**. NDF-REINFORCE is based on REINFORCE algorithm  (Williams, 1992), an elegant Monto-Carlo based policy gradient method which favors action with high sampled reward. The algorithm details are listed in Algorithm 2. Particularly, as indicated in equation 3, NDF-REINFORCE will support data filtration policy leading to higher cumulative reward $v_t$.

---

**Algorithm 2** NDF-REINFORCE algorithm to train NDF policy.

---

**Input**: Training data $D'$. Episode number $L$. Mini-batch size $M$. Discount factor $\gamma \in [0, 1]$.
**for** each episode $l = 1, 2, \cdots, L$ **do**
 Initialize the base neural network model.
 Shuffle $D'$ to get the mini-batches sequence $D' = \{D_1, D_2, \cdots, D_T\}$.
 **for** $t = 1, \cdots, T$ **do**
 Sample data filtration action for each data instance in $D_t = \{d_1, \cdots, d_M\}$: $a = \{a_m\}_{m=1}^M, a_m \propto A(s_m, a; \Theta)$, $s_m$ is the state corresponding to the $d_m$
 Update neural network model by Gradient Descent based on the selected data in $D_t$.
 Receive reward $r_t$.
 **end for**
 **for** $t = 1, \cdots, T$ **do**
 Compute cumulative reward $v_t = r_t + \gamma r_{t+1} + \cdots + \gamma^{T-t} r_T$.
 Update policy parameter $\Theta$:

$$\Theta \leftarrow \Theta + \alpha v_t \sum_m \frac{\partial \log A(s, a_m; \Theta)}{\partial \Theta} \qquad (3)$$

 **end for**
**end for**
**Output**: The NDF policy network $A(s, a; \Theta)$.

---

**NDF-ActorCritic**.

The gradient estimator in REINFORCE poses high variance given its Monto-Carlo nature. Furthermore, it is quite inefficient to update policy network only once in each episode. We therefore design NDF-ActorCritic algorithm based on value function estimation. In NDF-ActorCritic, a parametric value function estimator $Q(s, a; W)$ (i.e., a critic) with parameters $W$ for estimating state-action

---

[2]The margin for a training instance $(x, y)$ is defined as $P(y|x) - \max_{y' \neq y} P(y'|x)$ (Cortes et al., 2013)

value function is leveraged to avoid the high variance of $v_t$ from Monto-Carlo sampling in NDF-REINFORCE. It remains an open and challenging question that *how to define optimal value function estimator $Q(s, a; W)$ for SGD-MDP*. Particularly in this work, as a preliminary attempt, the following function is used as the critic:

$$Q(s, a; W) = \sigma(w_0^T relu(f(s)W_1 a) + b), \tag{4}$$

where $f(s) = (f(s_1); f(s_2); \cdots, f(s_M))$ is a matrix with $M$ rows and each row $f(s_m)$ represents state features for the corresponding training instance $d_m$. $W = \{w_0, W_1, b\}$ is the parameter set to be learnt by Temporal-Difference algorithm. Base on such a formulation, the details of NDF-ActorCritic is listed in Algorithm 3.

---

**Algorithm 3** NDF-ActorCritic algorithm to train NDF policy.

---

**Input**: Training data $D'$. Episode number $L$. Mini-batch size $M$. Discount factor $\gamma \in [0, 1]$.
**for** each episode $l = 1, 2, \cdots, L$ **do**
 Initialize the base neural network model.
 Shuffle $D'$ to get the mini-batches sequence $D' = \{D_1, D_2, \cdots, D_T\}$.
 **for** $t = 1, \cdots, T$ **do**
 Sample data filtration action for each data instance in $D_t = \{d_1, \cdots, d_M\}$: $a = \{a_m\}_{m=1}^M$, $a_m \propto A(s_m, a; \Theta)$, $s_m$ is the state corresponding to the $d_m$ . $s = \{s_m\}_{m=1}^M$.
 Update neural network model by Gradient Descent based on the selected data.
 Receive reward $r_t$.
 Update policy(actor) parameter $\Theta$: $\Theta \leftarrow \Theta + \alpha Q(s, a; W) \sum_m \frac{\partial \log A(s, a_m; \Theta)}{\partial \Theta}$.
 Update critic parameter $W$:

$$q = r_{t-1} + \gamma Q(s, a; W) - Q(s', a'; W), \quad W = W - \beta q \frac{\partial Q(s', a'; W)}{\partial W} \tag{5}$$

 $a' \leftarrow a, s' \leftarrow s$
 **end for**
**end for**
**Output**: The NDF policy network $A(s, a; \Theta)$.

---

## 3 EXPERIMENTS

### 3.1 EXPERIMENTS SETUP

We conduct experiments on two different tasks/models: IMDB movie review sentiment classification (with Recurrent Neural Network) and MNIST digital image classification (with Multilayer Perceptron Network). Different data filtration strategies we applied to SGD training include:

- **Unfiltered SGD**. The SGD training algorithm without any data filtration. Here rather than vanilla sgd (c.f. equation 1), we use its advanced variants such as Adadelta (Zeiler, 2012) or Adam (Kingma & Ba, 2014) to each of the task.

- **Self-Paced Learning (SPL)** (Kumar et al., 2010). It refers to filtering training data by its 'hardness', as reflected by loss value. Mathematically speaking, those training data $d$ satisfying $l(d) > \eta$ will be filtered out, where the threshold $\eta$ grows from smaller to larger during training process.

  In our implementation, to improve the robustness of SPL, following the widely used trick (Jiang et al., 2014b), we filter data using its loss rank in one mini-batch, rather than the absolute loss value. That is to say, we filter data instances with top $K$ largest training losses within a $M$-sized mini-batch, where $K$ linearly drops from $M - 1$ to 0 during training.

- **NDF-REINFORCE**. The policy trained with NDF-REINFORCE, as shown in Algorithm 2.

  We use a signal to indicate training speed as reward. To be concrete, we set an accuracy threshold $\tau \in [0, 1]$ and record the first mini-batch index $i_\tau$ in which validation accuracy

exceeds $\tau$, then the reward is set as $r_T = -\log(\tau/T)$. Note here only terminal reward exists (i.e., $r_t = 0, \forall t < T$).

- **NDF-ActorCritic**. The policy trained with NDF-ActorCritic, as shown in Algorithm 3. Discount factor is set as $\gamma = 0.95$.

  Since actor-critic algorithm makes it possible to update policy per time step, rather than per episode, different with the terminal reward set in NDF-REINFORCE, validation accuracy is used as the immediate reward for each time step. To save time cost, only part of validation set is extracted to compute validation accuracy.

- **Randomly Drop.** To conduct more comprehensive comparison, for NDF-REINFORCE and NDF-ActorCritic, we record the ratio of filtered data instances per epoch, and then randomly filter data in each mini-batch according to the logged ratio. In this way we form two more baselines, referred to as RandDropREINFORCE and RandDropActorCritic respectively.

For all strategies other than Plain SGD, we make sure that the base neural network model will not be updated until $M$ un-trained, yet selected data instances are accumulated. In that way we make sure that the batch size are the same for every strategies (i.e., $M$), thus convergence speed is only determined by the effectiveness of data filtration strategies, not by different batch size led by different number of filtered data. For NDF strategies, we initialize $b = 2$ (c.f. equation 2), with the goal of maintaining training data at the early age, and use Adam (Kingma & Ba, 2014) to optimize the policy. The model is implemented with Theano (Theano Development Team, 2016) and run on one Telsa K40 GPU.

## 3.2 IMDB SENTIMENT CLASSIFICATION

**IMDB movie review dataset**[3] is a binary sentiment classification dataset consisting of $50k$ movie review comments with positive/negative sentiment labels (Maas et al., 2011). We apply LSTM (Hochreiter & Schmidhuber, 1997) RNN to each sentence, and the last hidden state of LSTM is fed into a logistic regression classifier to predict the sentiment label (Dai & Le, 2015). The model size (i.e., word embedding size $\times$ hidden state size) is $256 \times 512$ and mini-batch size is set as $M = 16$. Adadelta (Zeiler, 2012) is used to perform LSTM model training.

The IMDB dataset contains $25k$ training sentences and $25k$ test sentences. For NDF-REINFORCE and NDF-ActorCritic, from all the training data we randomly sample $10k$ and $5k$ as the training/validation set to learn data filtration policy. For NDF-REINFORCE, the validation accuracy threshold is set as $\tau = 0.8$. For NDF-ActorCritic, the size of sub validation set to compute immediate reward is $1k$. The episode number is set as $L = 30$. Early stop on validation set is used to control training process in each episode.

The detailed results are shown in Figure 2, whose $x$-axis represents the number of effective training instances and $y$-axis denotes the accuracy on test dataset. All the curves are results of $5$ repeated runs. From the figure we have the following observations:

- NDF (shown by the two solid lines) significantly boosts the convergence of SGD training for LSTM. With much less data, NDF achieves satisfactory classification accuracy. For example, NDF-REINFORCE achieves $80\%$ test accuracy with only roughly half training data (about $40k$) of Plain SGD consumes (about $80k$). Furthermore, NDF significantly outperforms the two Randomly Drop baselines, demonstrating the effectiveness of learnt policies.

- Self-Paced Learning (shown by the red dashed line) helps for the initialization of LSTM, however, it delays training after the middle phrase.

- For the two variants of NDF, NDF-REINFORCE performs better than NDF-ActorCritic. Our conjecture for the reason is: 1) For NDF-REINFORCE, we use a terminal reward fully devoted to indicate training convergence; 2) The critic function (c.f., equation 4) may not be expressive enough to approximate true state-action value functions. Deep critic function should be the next step.

---

[3]http://ai.stanford.edu/~amaas/data/sentiment/

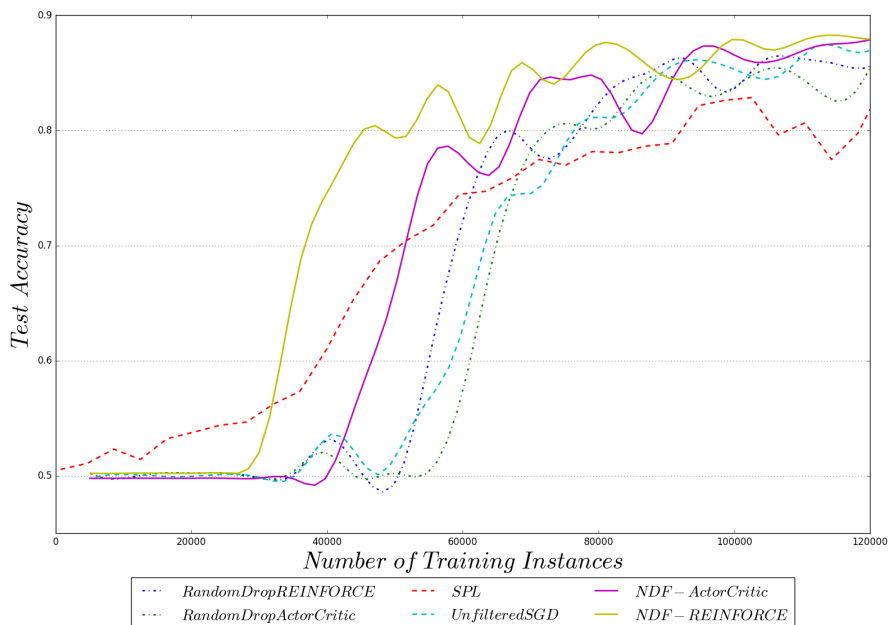

Figure 2: Test accuracy curves of different data filtration strategies on IMDB sentiment classification dataset. The $x$-axis records the number of effective training instances.

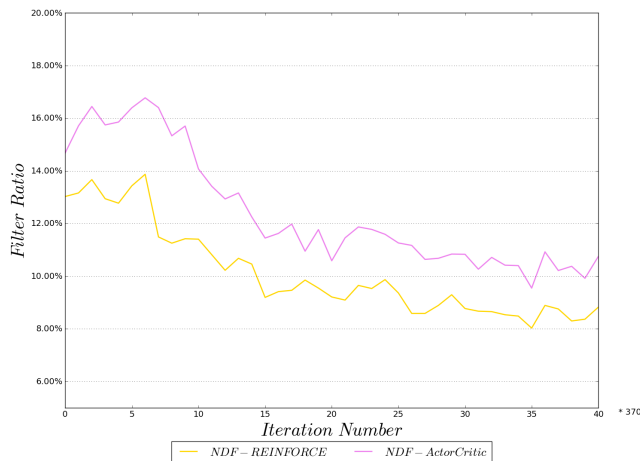

Figure 3: Data filtration ratio during training LSTM with NDF-REINFORCE and NDF-ActorCritic policies.

To better understand the learnt policies of NDF, in Figure 3 we plot the ratio of filtered data instances per every certain number of iterations. It can be observed that more and more training data are kept during the training process, which are consistent with the intuition of Curriculum Learning and Self-Paced Learning. Furthermore, the learnt feature weights for NDF policies (i.e. $\theta$ in equation 2) are listed in Table 1. From the table, we can observe:

- Longer movie reviews, with positive sentiments are likely to be kept.

- Margin plays critical value in determining the importance of data. As reflected by its fairly large positive weights, training data with large margin is likely to be kept.

- Note that the feature $-\log p_y$ is the training loss, its negative weights mean that training instances with larger loss values tend to be filtered, thus more and more data will be kept since loss values get smaller and smaller during training, which is consistent with the curves

| Feature | $y_0$ | $y_1$ | normalized sentence length | average historical training accuracy | normalized iteration | $\log p_0$ | $\log p_1$ | $-\log p_y$ | margin | bias $b$ |
|---|---|---|---|---|---|---|---|---|---|---|
| NDF -REINFORCE | -0.03 | 0.82 | 0.12 | -0.11 | -0.53 | 0.26 | 0.06 | -0.22 | 1.10 | 2.18 |
| NDF -ActorCritic | -0.08 | 0.77 | 0.20 | -0.13 | -0.61 | 0.20 | 0.04 | -0.12 | 1.12 | 1.84 |

Table 1: Feature weights learnt for NDF policies learnt in IMDB sentiment classification. The first row lists all the features (i.e., $f(s)$) categorized into the three classes described in Section 2. *normalized* means the feature value is scaled between $[0, 1]$. $[y_0, y_1]$ is the 1-of-2 representation for sentiment label.

in Figure 3. However, such a trend is diminished by the negative weight values for neural network features, i.e., historical training accuracy and normalized iteration.

## 3.3 IMAGE CLASSIFICATION ON CORRUPTED-MNIST

We further test different data filtration strategies for multilayer perceptron network training on image recognition task. The dataset we used is MNIST, which consists of $60k$ training and $10k$ testing images of handwritten digits from 10 categories (i.e., $0, \cdots, 9$). To further demonstrate the effectiveness of the proposed neural data filter in automatically choosing important instances for training, we manually corrupt the original MNIST dataset by injecting some noises to the original pictures as follows: We randomly split $60k$ training images into ten folds, and flip $(i-1) \times 10\%$ randomly chosen pixels of each image in the $i$-th fold, $i = 1, 2, \cdots, 10$. The $10k$ test set are remained unchanged. Flipping a pixel means setting its value $r$ as $r = 1.0 - r$. Such a corrupted dataset is named as *C-MNIST*. Some sampled images from C-MNIST are shown in Figure 4.

A three-layer feedforward neural network with size $784 \times 300 \times 10$ is used to classify the C-MNIST dataset. For data filtration policy, different from the single-layer logistic regression in equation 2, in this task, NDF-REINFORCE and NDF-ActorCritic leverage a three-layer neural network with model size $24 \times 12 \times 1$ as policy network, where the first layer node number $24$ is the dimension of state features $f_s$ [4], and sigmoid function is used as the activation function for the middle layer. $10k$ randomly selected images out of $60k$ training set acts as validation set to provide reward signals to NDF-REINFORCE and NDF-ActorCritic. For NDF-REINFORCE, the validation accuracy threshold is set as $\tau = 0.90$. For NDF-ActorCritic, the immediate reward is computed on the whole validation set. The episode number for policy training is set as $L = 50$ and we control training in each episode by early stopping based on validation set accuracy. We use Adam (Kingma & Ba, 2014) to optimize policy network.

The test set accuracy curves (averaged over five repeated runs) of different data filtration strategies are demonstrated in Figure 5. From Figure 5 we can observe:

- Similar to the result in IMDB sentiment classification, NDF-REINFORCE achieves the best convergence speed;
- The performance of NDF-ActorCritic is inferior to NDF-REINFORCE. In fact, NDF-ActorCritic acts similar to sgd training without any data filtration. This further shows although Actor-Critic reduces variance compared with REINFORCE, the difficulty in designing/training better critic functions hurts its performance.

## 4 RELATED WORK

Plenty of previous works talk about data scheduling (e.g., filtration and ordering) strategies for machine learning. A remarkable example is Curriculum Learning (CL) (Bengio et al., 2009) showing that a data order from *easy* instances to *hard* ones, a.k.a., a *curriculum*, benefits learning process.

---

[4] $f_s$ is similar to the features in Table 1, except that $(y_0, y_1)$ and $(\log p_0, \log p_1)$ are switched into $(y_0, \cdots, y_9)$ and $(\log p_0, \cdots, \log p_9)$ respectively, given there are ten target classes in mnist classification.

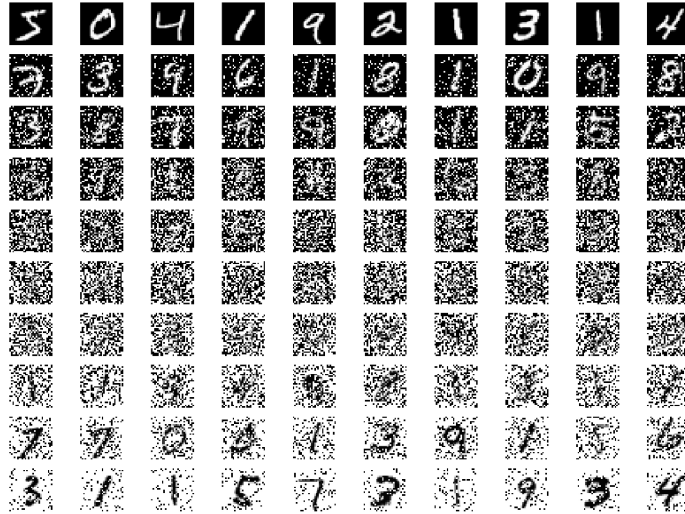

Figure 4: Sampled pictures from C-MNIST dataset. Each row represents a corrupted fold in training set, with the percentage of flipped pixels growing from $0\%$ (top row) to $90\%$ (bottom row).

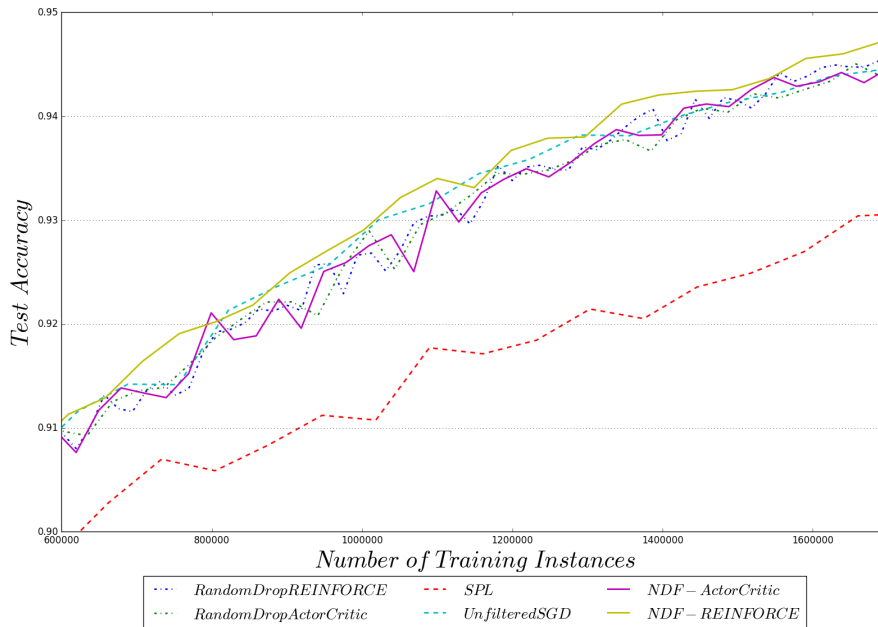

Figure 5: Test accuracy curves of different data filtration strategies on C-MNIST dataset. The $x$-axis records the number of effective training instances.

The measure of *hardness* in CL is typically determined by heuristic understandings of data (Bengio et al., 2009; Spitkovsky et al., 2010; Tsvetkov et al., 2016). As a comparison, Self-Paced Learning (SPL) (Kumar et al., 2010; Jiang et al., 2014a;b; Supancic & Ramanan, 2013) quantifies the *hardness* by the loss on data. In SPL, those training instances with loss values larger than a threshold $\eta$ will be neglected and $\eta$ gradually increases in the training process such that finally all training instances will play effects. Apparently SPL can be viewed as a data filtration strategy considered in this paper.

Recently researchers have noticed the importance of data scheduling for training Deep Neural Network models. For example, in (Loshchilov & Hutter, 2015), a simple batch selection strategy based on the loss values of training data is proposed for speed up neural networks training. (Tsvetkov et al., 2016) leverages Bayesian Optimization to optimize a curriculum function for training distributed word representations. The authors of (Sachan & Xing, 2016) investigated several hand-crafted criteria for data ordering in solving Question Answering tasks based on DNN. Our works differs significantly with these works in that 1) We aim to filter data in randomly arrived mini-batches in training process to save computational efforts, rather than actively select mini-batch; 2) We leverage reinforcement learning to automatically derive the optimal policy according to the feedback of training process, rather than use naive and heuristic rules.

The proposed Neural Data Filter (NDL) for data filtration is based on deep reinforcement learning (DRL) (Mnih et al., 2013; 2016; Lillicrap et al., 2015a; Silver et al., 2016), which applies deep neural networks to reinforcement learning (Sutton & Barto, 1998). In particular, NDL belongs to policy based reinforcement learning, seeking to search directly for optimal control policy. REINFORCE (Williams, 1992) and actor-critic (Konda & Tsitsiklis, 1999) are two representative policy gradient algorithms, with the difference that actor-critic adopts value function approximation to reduce the high variance of policy gradient estimator in REINFORCE.

## 5 CONCLUSION

In this paper we introduce Neural Data Filter (NDF), a reinforcement learning framework to select/filter data for training deep neural network. Experiments on several deep neural networks training demonstrate that NDF boosts the convergence of Stochastic Gradient Descent. Going beyond data filtration, the proposed framework is able to supervise any sequential training process, thus opens a new view for self-adaptively tuning/controlling machine learning process.

As to future work, on one aspect, we aim to test NDF to more tasks and models, such as Convolutional Neural Network (CNN) for image classification. We would also plan to give clearer explanation on the behavior of NDF, such as what data is dropped at different phrases of training, and whether the proposed critic function is good enough. On the other aspect, we aim to apply such a reinforcement learning based *teacher-student* framework to other strategy design problems for machine learning, such as hyper-parameter tuning, structure learning and distributed scheduling, with the hope of providing better guidance for controlled training process.

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
