# Peer review of "Neural Data Filter for Bootstrapping Stochastic Gradient Descent"

_ICLR 2017 — rejected_

[Official Review · AnonReviewer1 · rating 6 · confidence 4 · 16 Dec 2016 (modified: 19 Jan 2017)]

This work proposes to augment normal gradient descent algorithms with a "Data Filter", that acts as a curriculum teacher by selecting which examples the trained target network should see to learn optimally. Such a filter is learned simultaneously to the target network, and trained via Reinforcement Learning algorithms receiving rewards based on the state of training with respect to some pseudo-validation set.


Stylistic comment, please use the more common style of "(Author, year)" rather than "Author (year)" when the Author is *not* referred to or used in the sentence.
E.g. "and its variants such as Adagrad Duchi et al. (2011)" should be "such as Adagrad (Duchi et al., 2011)", and  "proposed in Andrychowicz et al. (2016)," should remain so.

I think the paragraph containing "What we need to do is, after seeing the mini-batch Dt of M training instances, we dynamically determine which instances in Dt are used for training and which are filtered." should be clarified. What is "seeing"? That is, you should mention explicitly that you do the forward-pass first, then compute features from that, and then decide for which examples to perform the backwards pass.


There are a few choices in this work which I do not understand:

Why wait until the end of the episode to update your reinforce policy (algorithm 2), but train your actor critic at each step (algorithm 3)? You say REINFORCE has high variance, which is true, but does not mean it cannot be trained at each step (unless you have some experiments that suggest otherwise, and if so they should be included or mentionned in the paper).

Similarly, why not train REINFORCE with the same reward as your Actor-Critic model? And vice-versa? You claim several times that a limitation of REINFORCE is that you need to wait for the episode to be over, but considering your data is i.i.d., you can make your episode be anything from a single training step, one D_t, to the whole multi-epoch training procedure.


I have a few qualms with the experimental setting:
- is Figure 2 obtained from a single (i.e. one per setup) experiment? From different initial weights? If so, there is no proper way of knowing whether results are chance or not! This is a serious concern for me.
- with most state-of-the-art work using optimization methods such as Adam and RMSProp, is it surprising that they were not experimented with.
- it is not clear what the learning rates are; how fast should the RL part adapt to the SL part? Its not clear that this was experimented with at all.
- the environment, i.e. the target network being trained, is not stationnary at all. It would have been interesting to measure how much the policy changes as a function of time. Figure 3, could both be the result of the policy adapting, or of the policy remaining fixed and the features changing (which could indicate a failure of the policy to adapt).
- in fact it is not really adressed in the paper that the environment is non-stationary, given the current setup, the distribution of features will change as the target network progresses. This has an impact on optimization.
- how is the "pseudo-validation" data, target to the policy, chosen? It should be a subset of the training data. The second paragraph of section 3.2 suggests something of the sort, but then your algorithms suggest that the same data is used to train both the policies and the networks, so I am unsure of which is what.


Overall the idea is novel and interesting, the paper is well written for the most part, but the methodology has some flaws. Clearer explanations and either more justification of the experimental choices or more experiments are needed to make this paper complete. Unless the authors convince me otherwise, I think it would be worth waiting for more experiments and submitting a very strong paper rather than presenting this (potentially powerful!) idea with weak results.

[Official Review · AnonReviewer3 · rating 4 · confidence 5 · 19 Dec 2016 (modified: 23 Jan 2017)]

Final review: The writers were very responsive and I agree the reviewer2 that their experimental setup is not wrong after all and increased the score by one.  But I still think there is lack of experiments and the results are not conclusive. As a reader I am interested in two things, either getting a new insight and understanding something better, or learn a method for a better performance. This paper falls in the category two, but fails to prove it with more throughout and rigorous experiments. In summary the paper lacks experiments and results are inconclusive and I do not believe the proposed method would be quite useful and hence not a conference level publication. 

--
The paper proposes to train a policy network along the main network for selecting subset of data during training for achieving faster convergence with less data.

Pros:
It's well written and straightforward to follow
The algorithm has been explained clearly.

Cons:
Section 2 mentions that the validation accuracy is used as one of the feature vectors for training the NDF. This invalidates the experiments, as the training procedure is using some data from the validation set.

Only one dataset has been tested on. Papers such as this one that claim faster convergence rate should be tested on multiple datasets and network architectures to show consistency of results. Especially larger datasets as the proposed methods is going to use less training data at each iteration, it has to be shown in much larger scaler datasets such as Imagenet.

As discussed more in detail in the pre-reviews question, if the paper is claiming faster convergence then it has to compare the learning curves with other baselines such Adam. Plain SGD is very unfair comparison as it is almost never used in practice. And this is regardless of what is the black box optimizer they use. The case could be that Adam alone as black box optimizer works as well or better than Adam as black box + NDF.

[Reviewer Comment · AnonReviewer2 · rating 7 · 20 Dec 2016]
**data filtering for faster sgd**

Paper is easy to follow, Idea is pretty clear and makes sense.
Experimental results are hard to judge, it would be nice to have other baselines.
For faster training convergence, the question is how well tuned SGD is, I didn't
see any mentioning of learning rate schedule. Also, it would be important to test
this on other data sets. Success with filtering training data could be task dependent.

[Author Response · Fei Tian · 13 Jan 2017]
**Reply for reviewer2**

Dear Reviewer, 

Thank you very much for your positive comments and scores!  As stated before, we are using Adadelta as the basic optimizer, rather than simple SGD. Very sorry for the misunderstanding brought by the term `Plain SGD’ in the paper. It just means `Adadelta without any data filtration'. In addition we are working on new datasets and conducting more qualitative analysis.

[Author Response · Fei Tian · 20 Jan 2017]
**A New Version**

Dear All Reviewers,

We have updated a new version, which: 1) adds a new experiment on mnist dataset to further verify the effectiveness of our proposed NDF algorithm; 2) makes a more dedicated and clearer description towards baseline method, REINFORCE v.s. Actor-Critic and validation setup.

We hope the new version can remove your previous concerns towards this work, even it is approaching the end of rebuttal period (we feel it sorry to submit it late).

Thanks again for all of your valuable comments and suggestions to this work!

Best,
Fei

[Final Decision · Program Chairs · 06 Feb 2017]
**ICLR committee final decision**

The authors propose a meta-learning algorithm which uses an RL agent to selectively filter training examples in order to maximise a validation loss. There was a lot of discussion about proper training/validation/test set practices. The author's setup seems to be correct, but the experiments are quite limited. Pro - interesting idea, very relevant for ICLR. Con - insufficient experiments. This is a cool idea which could be a nice workshop contribution.